# Identification of Tomato Ve1 Homologous Proteins in Flax and Assessment for Race-Specific Resistance in Two Fiber FlaxCultivars against *Verticillium dahliae* Race 1

**DOI:** 10.3390/plants10010162

**Published:** 2021-01-15

**Authors:** Adrien Blum, Lisa Castel, Isabelle Trinsoutrot-Gattin, Azeddine Driouich, Karine Laval

**Affiliations:** 1UNILASALLE—Campus Rouen, Aghyle Unit, SFR Normandie Végétal FED 4277, 3 rue du Tronquet, 76134 Mont-Saint-Aignan Cedex, France; Lisa.CASTEL@unilasalle.fr (L.C.); Isabelle.GATTIN@unilasalle.fr (I.T.-G.); Karine.LAVAL@unilasalle.fr (K.L.); 2Glycobiologie et Matrice Extracellulaire Végétale EA 4358, SFR Normandie Végétal FED 4277, Université de Rouen, 76821 Mont-Saint-Aignan, France; azeddine.driouich@univ-rouen.fr

**Keywords:** *Verticillium dahliae*, race specific resistance, fiber flax

## Abstract

In the last decade, the soil borne fungal pathogen *Verticillium dahliae* has had an increasingly strong effect on fiber flax (*Linum usitatissimum* L.), thus causing important yield losses in Normandy, France. Race-specific resistance against *V. dahliae* race 1 is determined by tomato Ve1, a leucine-rich repeat (LRR) receptor-like protein (RLP). Furthermore, homologous proteins have been found in various plant families. Herein, four homologs of tomato Ve1 were identified in the flax proteome database. The selected proteins were named LuVe11, LuVe12, LuVe13 and LuVe14 and were compared to other Ve1. Sequence alignments and phylogenic analysis were conducted and detected a high similarity in the content of amino acids and that of the *Verticillium* spp. race 1 resistance protein cluster. Annotations on the primary structure of these homologs reveal several features of tomato Ve1, including numerous copies of a 28 amino acids consensus motif [XXIXNLXXLXXLXLSXNXLSGXIP] in the LRR domain. An in vivo assay was performed using *V. dahliae* race 1 on susceptible and tolerant fiber flax cultivars. Despite the presence of homologous genes and the stronger expression of *LuVe11* compared to controls, both cultivars exhibited symptoms and the pathogen was observed within the stem. Amino acid substitutions within the segments of the LRR domain could likely affect the ligand binding and thus the race-specific resistance. The results of this study indicate that complex approaches including pathogenicity tests, microscopic observations and gene expression should be implemented for assessing race-specific resistance mediated by Ve1 within the large collection of flax genotypes.

## 1. Introduction

Flax (*Linum usitatissimum* L.) is an ancient crop which is currently widely cultivated as a source of fiber, oil and medicine-related compounds [1]. According to [2], *L. usitatissimum* convar. *elongatum*, a.k.a. fiber flax, incorporates cultivars for fiber that is used in industries such as textile manufacture and composite materials [3]. In 2014, more than 67,000 ha of fiber flax were planted in France, which is the leading country in fiber flax production (FAOSTAT 2014). More than 50% of the land used for French flax cultivation is located in the region of Normandy because it provides an appropriate pedo-climatic environment for the cultivation of fiber flax (http://www.chambre-agriculture-normandie.fr/panorama-lin-normandie/). The importance of this crop has led to whole-genome sequencing of the cultivar CDC Bethune, a linseed cultivar growing on the majority of flax acreage in Canada [4]; this provides substantial resources for the predicted proteome [5].

Previous research on flax diseases has focused on fusarium wilt (*Fusarium oxysporum* f.sp. *lini*) and rust (*Melampsora lini*), which are the main limiting factors in flax production [6]. According to French monitoring organizations, Verticillium wilt, a vascular disease has become a substantial problem for fiber flax cultivation, and it has caused important yield losses during the last decade [7,8]; http://www.fiches.arvalis-infos.fr/). Verticillium wilt is caused by the soil borne fungus *Verticillium dahliae* Kleb. and is a major threat to over 200 plant species including a large range of plant crops [9,10,11]. This devastating disease causes significant yield losses and leads to substantial economic losses in different regions of the world [11]. Generally, symptoms begin to appear a few weeks after inoculation, for example, it takes eight to ten weeks for lettuce, fourteen days for cotton and two weeks for *Arabidopsis* to start wilting [12,13,14]. Infection commonly causes development of irregular chlorotic patches that turn necrotic on leaves. In the oldest shoots, wilting symptoms start in one half of an infected leaf and progress acropetally, from the base to the apex [15].

Since the prohibition of methyl bromide fumigation due to environmental problems, control of Verticillium wilt is challenging because no effective solutions exist. However, research focused on varietal selection, agronomic methods and antagonist microbes have opened the way to promising alternatives [15,16,17]. Incompatible interaction with *V. dahliae* race 1 pathotypes was detected on several tomato cultivars, thereby preventing disease progression on the stem and leaves of plants [18,19]. *V. dahliae* race 1 strains are more aggressive than race 2 strains on susceptible tomato cultivars [20,21]. Tomato race-specific resistance is related to an allelic form of *Ve1*, a gene which encodes a cell-surface glycoprotein [22,23,24]. Accordingly, *V. dahliae* race 1 encodes a secreted effector Ave1 involved in race-specific resistance and which contributes to fungal virulence [25]. Race-specific resistance was observed in the *Solanum* genus [26,27,28,29] and in other plant cultivars such as lettuce [30] and cotton [31,32], which suggests that a common interfamily race-specific resistance exists.

Tomato *Ve1* encodes a membrane receptor that belongs to the receptor-like protein (RLP) class. The RLP class encompasses proteins that often play a role in disease resistance (Wang et al., 2008). Among the RLP resistance proteins, Cf-4 and Cf-9 confer resistance to tomato leaf mold caused by *Cladosporium fulvum* [33,34]; HcrVf confers resistance to apple scab disease caused by *Venturia inaequalis* [35,36,37]; and, as discovered more recently, LepR3 confers resistance to *Leptosphaeria maculans* on rapeseed [38]. Analysis of the primary structure of Ve1 shows five distinct domains: domains A-E. N-terminus domain A is a signal peptide comprised of hydrophobic amino acids and a copy of leucine zipper-like motif LX(6)LX(6)LX(6)L, which facilitates protein dimerization. Domain B (the so-called eLRR domain) is an extracellular LRR formed by 37 imperfect copies of a 28 amino acids consensus motif [XXIXNLXXLXXLXLSXNXLSGXIP], which are involved in protein–protein interaction in plants [33] and a non LRR segment. This domain also has 28 N-glycosylation sites identified by the NX[ST] motif. Domain C is extracytosolic and negatively charged. As frequently observed in membrane proteins, domain D is a transmembrane segment formed by hydrophobic amino acids. In C-terminus, domain E is cytosolic, positively charged and includes a copy of the mammal-specific endocytosis signal: [ED]XXXφ or YXXφ, where φ is a hydrophobic amino acid [23]. Generally, RLPs contain a short cytosolic segment, the function of which remains unclear [39].

Since race-specific resistance to *V. dahliae* race 1 has been observed in various plant genera, the aim of this study was to investigate the potential of applying such a mechanism to flax. First, an in silico screening was performed to select the flax predicted proteome homologous proteins of tomato Ve1 (accession ACR33105.1); the selection was based on the identity of the amino acids, phylogeny and functional annotations. Second, flax homologous genes were assessed using an in vivo assay, which involved conducting PCR, RT-qPCR, a pathogenicity test and microscopic observations. Plant behavior upon infection was analyzed using a GFP-tagged *V. dahliae* race 1.

## 2. Methods

### 2.1. Ve1 Homologous Proteins Screening, Phylogeny and Primary Structure Analysis

The accessions of proteins annotated “Verticillium wilt disease resistance protein” on Genbank were ACR33105.1 (Ve1, *Solanum lycopersicum*), ACJ61469.1 (GbVe, *Gossypium barbadense*), AAP20229.1 (SlVe1, *Solanum lycopersicoides*), AAQ82053.1 (StVe, *Solanum torvum*), AAT51733.1 (*Solanum aethiopicum*), ACB99683.1 (mVe1, *Mentha longifolia*), ACB99693.1 (*Mentha spicata*), AES86946.1 (*Medicago truncatula*) and BAD68095.1 (*Oryza sativa* Japonica Group). The homologous proteins of Ve1 (accession ACR33105.1) were searched in *L. usitatissimum* cv. CDC Bethune proteome using BLASTP on Phytozome (https://phytozome.jgi.doe.gov). A multi-sequence alignment analysis was performed using BLASTP (http://blast.ncbi.nlm.nih.gov) between Ve1 and proteins selected in flax and “Verticillium wilt disease resistance protein”. The identity of the ORF amino acids sequence was expressed as a %.

### 2.2. Ve1 Homologous Proteins Phylogeny

A phylogenic analysis of resistance-like proteins (RLPs) using the maximum-likelihood method was conducted using the Phylogeny.fr webservice [40,41], which incorporates MUSCLE (Appendix A, Appendix A), Gblocks, PhyML and TreeDyn. In addition to “Verticillium wilt disease resistance protein”, resistance RLPs accessions AAA65235.1 (Cf-9, *Solanum pimpinellifolium*), CAA05268.1 (Cf-4, *Solanum habrochaites*), AJG42091.1 (LEPR3, *Brassica napus*) AJG42090.1 (LEPR3, *Brassica juncea*), AJG42088.1 (LEPR3, *Brassica rapa),* and CAC40826.1 (HcrVf2, *Malus floribunda*) were added.

### 2.3. Ve1 Homologous Proteins Primary Structure Analysis

Two independent analyses of primary structure were performed on Ve1 and the four selected Ve1 homologs in flax. First, signal peptide, non-cytoplasmic, transmembrane and cytoplasmic regions were determined by the Phobius prediction [42]. Secondary structure predictions of LuVe11 and Ve1 were performed using PSIPRED (http://bioinf.cs.ucl.ac.uk/psipred) and are shown in the Appendix A, Appendix A. Second, a protein alignment analysis was performed using ClustalW2 (http://www.ebi.ac.uk/Tools/msa/clustalw2/). Then, the five domains A-E were determined based on [23] including the 37 imperfect copies of a 28 amino acids consensus motif [XXIXNLXXLXXLXLSXNXLSGXIP] and the non LRR segment. ScanProsite (http://prosite.expasy.org/scanprosite/) was used to define protein patterns such as the N-glycosylation pattern N-X-[ST], Leucine Zipper-like pattern LX(6)LX(6)LX(6)L and the endocytosis signal pattern [ED]XXXφ or YXXφ, where φ stands for the hydrophobic amino acid.

### 2.4. Plant Material

Fiber flax seeds (*Linum usitatissimum* L.) cv. Télïos and cv. Adélie, which are susceptible and tolerant, respectively, were provided by Terre de Lin, Saint-Pierre-le-vigier France. The seeds were sterilized by immersing them in 70% ethanol for two minutes. They were then rinsed in sterile water for five minutes and immersed in sodium hypochlorite 2.6% (v/v) for ten minutes. Finally, they were rinsed three times in sterile water before sowing.

### 2.5. Pathogen Strain

The department of Plant Pathology of the University of California, Salinas, USA provided *V. dahliae*, the strain VdLs16, previously described as race 1 [41] and transformed with the GFP gene. Fluorescence was checked using an epi-fluorescent microscope (DM1000, Leica, Germany), which uses a 50W Hg lamp (Leica, Germany). To obtain 10^6^ conidia per ml, strain was grown on potato dextrose (26.5 g l^−1^) with sterile water in an Erlenmeyer flask. It was then placed in the agitator (150 rpm) for fifteen days in the dark at 20 °C. Conidia were isolated in phosphate buffered saline (PBS) by crushing fungal mass in a 100 µm pore sieve and then counted using the Malassez counting chamber.

### 2.6. Inoculation Procedure and Plants Cultivation

This research used the soil-drench method/peat plant cultivation according to Blum et al., (2018). Briefly, sterile seeds were sowed in Jiffy-7^®^ pots (Jiffy Products International BV, Moerdijk, The Netherlands) at the rate of 1 seed per pot. The bottom of the Jiffy pot contained ten-day old seedlings that were drenched for one hour with a conidia solution with a concentration of 10^6^ conidia per ml or in PBS for controls. This inoculation process was repeated a second time. The Jiffy pots were then transplanted into larger peat pots. After three weeks, plant stakes were set up. This assay was performed in a growth chamber at 23 °C, with a photoperiod consisting of 16 h of light and 8 h of darkness.

### 2.7. Symptoms and Pathogen Observation in Plants

Wilting symptoms were detected four weeks after the inoculation. For pathogen detection in plant tissues, samples were fixed in methanol and then stored at 4 °C [43]. All samples were mounted on a glass slide with distilled water. Pathogen fluorescence in plant tissues was detected using an epi-fluorescent microscope (DM1000, Leica, Germany) fitted with a 50W Hg lamp (Leica, Germany).

### 2.8. Plant DNA Extraction and Quantification

Up to 50 mg of plant root tissue was transferred and stored at −20 °C. A total DNA extraction was performed using the PowerPlant DNA Isolation kit (MoBio Laboratories, Carlsbad, CA, USA) following the manufacturer’s instructions. Total DNA concentrations were assessed using the Fluorescent DNA quantification kit (Bio-Rad, Hercules CA, USA) and Varioskan™ Flash Multimode Reader (ThermoScientific, Waltham, MA, USA) following the manufacturer’s instructions. All DNA samples were stored at −20 °C. 

### 2.9. Plant RNA Extraction, Quality Control and cDNA Synthesis

Three samples per date (J+0 corresponding to four hours post inoculation, D + 1, D + 3, D + 7 and D + 10) and per condition (control or inoculated by VdLs16) were analyzed. Up to 100 mg of plant root including controls and infected samples were quickly transferred to liquid nitrogen, and then crushed with a pestle. The total RNA extraction was performed using the RNeasy Plant Mini Kit (Qiagen, Hilden, Germany) following the manufacturer’s instructions. Samples were treated using the RNase-Free DNase Set (Qiagen, Hilden, Germany) during extraction and the TURBO DNA-*free*™ Kit (ThermoScientific, Waltham, MA, USA) after extraction. Total RNA concentrations and purity were assessed using the Experion™ Automated Electrophoresis System (Bio Rad, Hercules, CA, USA) following the manufacturer’s instructions. Forty µg of total RNA were reverse transcribed using the SuperScript^®^ VILO™ Master Mix (ThermoScientific, Waltham, MA, USA) and following the manufacturer’s instructions. All cDNA samples were stored at −20 °C.

### 2.10. Detection of the Ve1 Homologous Genes Using PCR

*Ve1* homologous genes detection was performed on fiber flax cv. Adélie using a set of primers described in Table 1. The primers were designed by the Primer3 software [44,45] and the Lus10003389.g, Lus10042239.g, Lus10026415.g and Lus10003387.g sequences were used as the template. The sequences were provided by the *L. usitatissimum* cv. CDC Bethune sequenced genome (Wang et al., 2012). *EF1A* and *UBI2* were amplified using LuEF1A_1F/LuEF1A_1R and LuUBI2_1F/LuUBI2_1R, respectively, [46] as positive control. End-point PCR was performed on the 25 µL reaction volume, which included 50 ng of flax DNA, 0.5 µM of each primer and 12.5 µL of the GoTaq^®^ Green Master Mix (Promega, Madison, WI, USA). PCR cycling was achieved through the GeneAmp^®^ PCR System 9700 (Applied Biosystems, Foster City, CA, USA) following the program: 95 °C for 2 min; 35 cycles of 95 °C for 45 s for 60 °C; 1 min; 72 °C for 30 s; 72 °C for 7 min; and finally, 4 °C for cooling. PCR products were visualized by electrophoresis on a 1% agarose gel stained by ethidium bromide.

### 2.11. LuVe11 Relative Expression by Real-Time PCR

The primers used in real-time PCR were LuVe11_1F/LuVe11_1R, which targeted *LuVe11* as “the target gene”. LuEF1A_1F/LuEF1A_1R and LuUBI2_1F/LuUBI2_1R were selected by [46] (Table 1) and targeted *EF1A* and *UBI2,* respectively, as “the housekeeping genes”. The final procedure on the 25 μL reaction volume contained 4 ng of cDNA extracted from the plant, 0.4 μM of each primer and 1 X LightCycler^®^ 480 DNA SYBR Green I Master mix (Roche Basel, Switzerland). Amplification was performed using a LightCycler 480 real-time PCR system (Roche, Basel, Switzerland). The PCR cycling conditions were 95 °C for 15 min and 50 cycles at 95 °C for 30 s, 60 °C for 30 s and 72 °C for 20 s. After the final amplification cycle, the melting curve profiles were obtained by heating the samples up to 95 °C, cooling them down to 60 °C and slowly heating them up again to 97 °C; this was done by gradually increasing the temperature by 1.1 °C every ten seconds while continuously measuring the fluorescence at 520 nm. Three-point standard curves of a three-fold dilution series (1:10 to 1:1000) comprised of a cDNA pool were used to calculate the PCR efficiency for each primer pair. The PCR efficiency and Cq were determined using the Roche second derivative maximum method. The real-time PCR reactions for the standard curve were repeated at least three times and PCR assays always included negative control (no DNA) and NRT negative controls (only RNA). Melting curves were used to detect if only one PCR product was amplified for each primer pair. The real-time PCR data was analyzed using the REST-MSC© version 2 [47,48]. Briefly, the software uses a method of Cq comparison and specifies if a significant difference exists between the control and the inoculated condition by using a non-parametric test, i.e., the pair fixed reallocation randomization test. Consequently, the target gene was determined to be upregulated or downregulated in the inoculated condition relative to the control.

## 3. Results

### 3.1. In Silico Analysis Reveals Ve1 Homologs in Flax

Genbank accession ACR33105.1 was used as the sequence query to search the predicted flax proteome for Ve1 homologs. Four candidate proteins were selected and named LuVe11 LuVe12, LuVe13 and LuVe14. The protein sizes were 1039 amino acids (aa) for LuVe11, 1090 aa for LuVe12, 1067 aa for LuVe13, and 1061 aa for LuVe14 (Table 2) whereas the Ve1 size was 1053 aa. Proteins annotated as “Verticillium wilt disease resistance” proteins on Genbank shared between 39.5% and 91.2% aa identity with Ve1 and belonged to the *Solanum* genus as well as other plant genera. LuVe11 shared 48.8% aa identity with Ve1 while LuVe12, LuVe13, and LuVe14 shared 45.3%, 45.4%, and 45.9%, respectively (Table 3). The phylogenic analysis reveals that Ve1, other “Verticillium wilt disease resistance” protein and the four flax homologs are clustered in a distinct group from the other disease resistant RLPs. The four flax homologs were closer to *Gossypium barbadense* GbVe than Ve1, and LuVe11 and LuVe14 were closer to GbVe than LuVe12 and LuVe13 (Figure 1). The phylogenic relationships between Ve1 homologs are shown in Appendix A, Appendix A, including the relationship between functional and non-functional homologs. This analysis suggests that duplications have occurred within genera and/or species. Furthermore, two independent analyses regarding the subcellular localization were performed, with functional annotations for being included in the second one.

A predictive analysis of Ve1 and the four flax proteins reveals that all sequences are comprised of a N-terminus peptide signal, a large non-cytosolic segment, a transmembrane segment and a small cytosolic segment in C-terminus except for LuVe14 which is comprised of a second transmembrane segment and a cytosolic segment in C-terminus (Appendix A). Furthermore, the five domains A-E according to [23] were investigated through a sequence alignment (Figure 2). Domain A matched a signal peptide, which was comprised of hydrophobic amino acids and a copy of a leucine zipper-like motif LX(6)LX(6)LX(6)L and was only found on Ve1. Domain B was an extracellular LRR containing 37 imperfect copies (B1 to B38, except for B32) of a 28 amino acids consensus motif [XXIXNLXXLXXLXLSXNXLSGXIP] and a non LRR segment (B32). LuVe11 contained 36 imperfect copies of the consensus motif (not B23), LuVe14 contained 34 copies (not B27, B28 and B29), and LuVe12 and LuVe13 contained 37 copies. This domain also included 28 N-glycosylation sites identified by the NX[ST] motif. Likewise, 27 potential N-glycosylation sites were found on LuVe11, 28 on LuVe14 and 23 on LuVe12 and LuVe13; several sites were in common and were conserved. A copy of the leucine zipper-like motif was identified in LuVe11 and LuVe13 in domain B. The following elements were present in all homologs: an extracytosolic and negatively charged in domain C, a transmembrane segment formed by hydrophobic amino acids in domain D, and in C-terminus, a cytosolic and positively charged in domain E and a copy of the mammal-specific endocytosis signal [ED]XXXφ or YXXφ, where φ stands for hydrophobic amino acid. A membrane-associated domain within domain E and hydrophobic amino acids were only found in LuVe14.

### 3.2. Disease Onset and Pathogen Colonization despite the Presence of Ve1 Homologous Genes and the Expression of LuVe11

The presence of *Ve1* homologs in fiber flax DNA (cv. Adélie) was investigated using a PCR assay. Generation of the primers was based on the cv. Béthune sequences. Three primer pairs were designed for each of the *Ve1* homologs *LuVe11*, *LuVe12*, *LuVe13* and *LuVe14.* Each primer pair and their respective amplicon sizes are compiled in Table 1. The PCR assay revealed the PCR products for each primer pair (Figure 3). Based on the molecular size marker, each PCR product matched its respective amplicons, which suggests that *LuVe11*, *LuVe12*, *LuVe13* and *LuVe14* are present in fiber flax DNA (cv. Adélie). The relative expression of *LuVe11* in the root tissues of control and inoculated conditions in the two fiber flax cultivars during the ten days post inoculation (dpi) were compared (Figure 4). According to the supplier, cv. Télïos is more susceptible to *V. dahliae* infection than cv. Adélie. Overall, *LuVe11* is up-regulated under inoculated conditions in all cultivars from four hpi (zero dpi) to ten dpi. The expression peak in cv. Télïos was detected at three dpi whilst in cv. Adélie, the strongest expression was detected right after inoculation. The pathogenicity test was combined with the microscopic observation of the stem and performed on two fiber flax cultivars (cv. Télïos and cv. Adélie) using the GFP-tagged *V. dahliae* race 1 strain. The first wilting symptoms were observed four weeks post inoculation (wpi) in both cultivars (Figure 5A,B,D,F). Plants exhibited necrosis on the apical part of their leaves and yellowing. Additionally, microscopic observations revealed *V. dahliae* hyphae within the stem of infected plants in both cultivars at four wpi(Figure 5C,E).

## 4. Discussion

The allelic *Ve1* gene confers race-specific resistance to *V. dahliae* race 1 on tomato plants [18,21,22,23]. *Ve1* encodes a leucine-rich repeat (LRR) receptor that belongs to the receptor-like proteins (RLPs) family [23,24]. The recent phylogenic analysis of *Ve1* homologs reveals that homologs are widely spread, even in phylogenetically distant land plants [49]. This study demonstrates the existence of *Ve1* homologs in an additional family (Linaceae) of flowering plants. Race-specific resistance governed by Ve1 homologs was also detected in various plant species including lettuce cultivars [30] and cotton cultivars [31,32], which suggests that a common interfamily resistance exists. Homologs of Ve1 (accession ACR33105.1) in flax were identified using BLASTP through flax predictive proteome. The identified proteins were subsequently named LuVe11, LuVe12, LuVe13 and LuVe14. The four flax homologs share between 45.3 and 48.8% amino acids identity to Ve1. Because LuVe11 has the largest amino acids identity, it was selected for further in vivo analysis. These proteins were annotated as “Verticillium wilt disease resistance” protein. Furthermore, a phylogenic analysis applying maximum-likelihood was conducted and revealed a *Verticillium* spp. race 1 resistance cluster that was distinct from other disease resistance RLPs. The prediction of subcellular localization reveals that flax Ve1 homologs have cell surface receptor features that are very similar to Ve1. Analysis performed by [4] revealed duplication of the whole-genome of the ancestor of *L. usitatissimum*, 5-9 million years ago. This duplication event could explain the presence of many Ve homologs in flax genome. Appendix A, Appendix A shows that the four proteins are clustered in a distinct clade that include GbVe but not Ve1. Moreover, these data showed that *LuVe11* and *LuVe14* are clustered together as well as *LuVe12* and *LuVe13*, indicating paralogy between the genes. Further investigations (data not shown) on genes orientation showed that *LuVe11* and *LuVe14* (also *LuVe12* and *LuVe13*) are oriented as tomato *Ve1* and *Ve2*.

The second analysis of domains A-E, in accordance with [23], identified the features of the domains in the flax Ve1 homologs. However, the features of each homolog differed, for example, LuVe14 has a larger C-terminus tail that contains a membrane-associated domain within domain E. Domain B forms the extracellular LRR, which is divided into 37 imperfect copies (B1 to B38, except B32) of a 28 amino acids consensus motif [XXIXNLXXLXXLXLSXNXLSGXIP] and a non LRR segment (B32). Domains containing LRR motifs are widespread in the plant kingdom and are involved in protein–protein interactions.

The variability of each LRR motif contributes to the ligand specificity, and consequently, to the resistance specificity [50]. Ref. [51] in each imperfect copy “eLRR” region, which corresponds in this study to the “B” region. Further studies confirm that tomato Ve1 segments eLRR30 to eLRR35 (B30 to B35) are crucial for triggering race-specific resistance whereas the first thirty eLRR contribute to ligand binding and could be replaced by the first thirty eLRR of non-functional tomatoVe2 [51]. In addition, Ref. [52] have proven that eLRR1 to eLRR8 and eLRR20 to eLRR 23 contribute to ligand binding and eLRR32 to eLRR37 are essential for triggering Ve1 function. Consequently, the substitution of amino acids from the segments eLRR1 to eLRR8 (B1 to B8), eLRR20 to eLRR23 (B20 to B23), and eLRR30 to eLRR37 (B30 to B37) to LuVe11, LuVe12, LuVe13 and LuVe14 can likely impact the ligand binding and the triggering of Ve1 resistance. Moreover, the substitutions of amino acids in LuVe11 compared to Ve1 imply secondary structure modifications (Appendix A, Appendix A) including within the crucial regions for ligand binding and resistance triggering. In these regions, 29 coils, 15 α-helix and ten β-strand were predicted in LuVe11 instead of 32 coils, 15 α-helix, and 15 β-strand in Ve1. This suggests that ligand binding proprieties should not be conserved. Nevertheless, this analysis is not able to identify secondary structure elements, which potentially render LuVe11 non-functional. The race-specific resistance failure observed in the in vitro test is likely related to this hypothesis. Thus, the interfamily resistance could be connected to the amino acid composition of these segments. Ultimately, this paper identified four eLRR-RLPs homologs of Ve1 tomato in the flax proteome cv. CDC Bethune. Amino acid sequences of the flax homologs reveal that they belong to the *Verticillium* spp. race 1 resistance protein group and indicated potential preserved functions. The genes that encode the four homologs (*LuVe11*, *LuVe12*, *LuVe13* and *LuVe14*) were identified in a fiber flax cultivar (cv. Adélie) that is usually planted in Normandy. Little is known about the downstream signaling of RLPs whereas there is much research on the downstream signaling of receptors like kinases (RLKs) [53,54,55]. However, the arrangement of the heterodimer, which combines RLPs and RLKs, was reported. For example, CLAVATA2 (CLV2) is supposed to interact with CLV1, a RLK which forms a heterodimer in order to bind the potential ligand [56,57,58]. Another report deals with the RLP flagellin sensing 2 (FLS2), which forms a heterodimer with the RLK receptor BRI associated to the kinase 1 (BAK1) in the presence of the ligand, the conserved protein flagellin flg22 [59,60]. Furthermore, B32 to B37 segments in Ve1 are considered to be a region of hetero-dimerization [52]. In this study, motifs (leucine zipper), which facilitate protein dimerization were detected in LuVe11 and LuVe13. These proteins interact and form heterodimers to bind the ligand. LuVe14 could be a good candidate because of its long C-terminus tail, which is likely involved in the intracellular signaling or as mentioned before, a RLK receptor member.

Nevertheless, the pathogenicity test and microscopic observations show that the disease is onset and pathogen colonization, thus cultivars (cv. Télïos and cv. Adélie) are susceptible when infected by a GFP-tagged *V. dahliae* race 1 strain until 4 wpi. From a phenotypic standpoint, resistance (i.e., no symptoms, stopping the pathogen progression), as described in the literature, was not observed in these two cultivars, which, however, have a different reaction to Verticillium wilt. However, a difference in a number of symptoms between the two cultivars was observed, and further symptomatic analysis would be necessary to determine the disease severity. The real-time PCR of *LuVe11* (gene whose protein is the closest to Ve1) relative expression suggests that, compared to the controls, the *LuVe11* is upregulated in the roots of both cultivars when inoculated. Together, these results show that race-specific resistance does not operate in the cultivars tested at 4 wpi. In reality, the presence of *Ve1* homologs does not guarantee the presence of an effective race-specific resistance such as StuVe2 (*Solanum tuberosum*), StoVe2 (*Solanum torvum*) or HLVe1-2B (*Humulus lupulus*) [49]. According to [24], the *Ve1* expression increases significantly in susceptible cultivars but only moderately in resistant cultivars because the fungus is stopped at an early stage of the infection. The expression data for *LuVe11* does not provide unique profiles that correspond to either the susceptible or the tolerant cultivar. Nevertheless, polymorphism in the *Ve1* coding sequence has been reported, including within resistant and susceptible tomato cultivars [24] and this preliminary report on flax using two cultivars, did not state that race-specific resistance does not operate within the genus *Linum*. Furthermore, polymorphism also exists in other RLPs. For example, a high nucleotide variation exists in *CLV2*, a gene encoding receptor-like proteins CLV2. These nucleotide variations were observed in the LRR coding sequence of *CLV2*, by comparing it with other genes and within different *Arabidopsis thaliana* ecotypes [61,62]. In this respect, it could be interesting to use flax diversity for assessing race-specific resistance mediated by Ve1 homologs and analyze *LuVe* nucleotide polymorphism. Actually, around 48,000 flax genotype accessions are distributed in 33 genebanks around the world and 10,000 could be unique [63]. Therefore, it is likely that one or more race-specific resistant flax genotypes can be found, as [18] observed in a small-fruited wild tomato from Peru. This study combined the pathogenicity test, microscopic observations and *LuVe11* relative expression, which provided an effective method for assessing the race-specific resistance of flax diversity.

## 5. Conclusions

*Ve1* homologs are widespread in land plants, including in the Linaceae family which encompasses flax. This suggests that race-specific resistance against *V. dahliae* race 1 could be present in flax. However, in vitro analysis failed to find this resistance. Candidate proteins in flax share many features with Ve1, but the amino acids substitutions in the LRR domain can have an impact on the ligand binding or resistance triggering. Nevertheless, this paper was able to develop a method for screening flax diversity and uncovering potential candidates.

## Figures and Tables

**Figure 1 plants-10-00162-f001:**
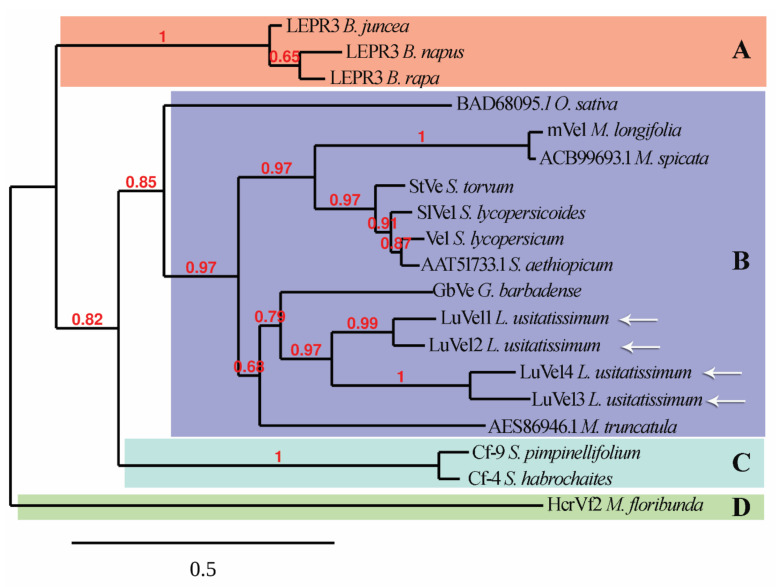
Phylogram showing evolutionary relationships between resistance receptor-like proteins (RLPs) in plants including candidate proteins in flax (arrows), determined by using the maximum-likelihood method. Branch lengths are proportional to evolutionary distances, branch values > 0.6 are given at the node to indicate the branch support. A indicates a Leptosphaeria maculans resistance RLPs cluster, B is relative to a Verticillium spp. race 1 resistance cluster, C to a Cladosporium fulvum cluster and cluster D to Venturia inaequalis resistance RLPs. Amino acids alignment is showed in Appendix A, Appendix A.

**Figure 2 plants-10-00162-f002:**
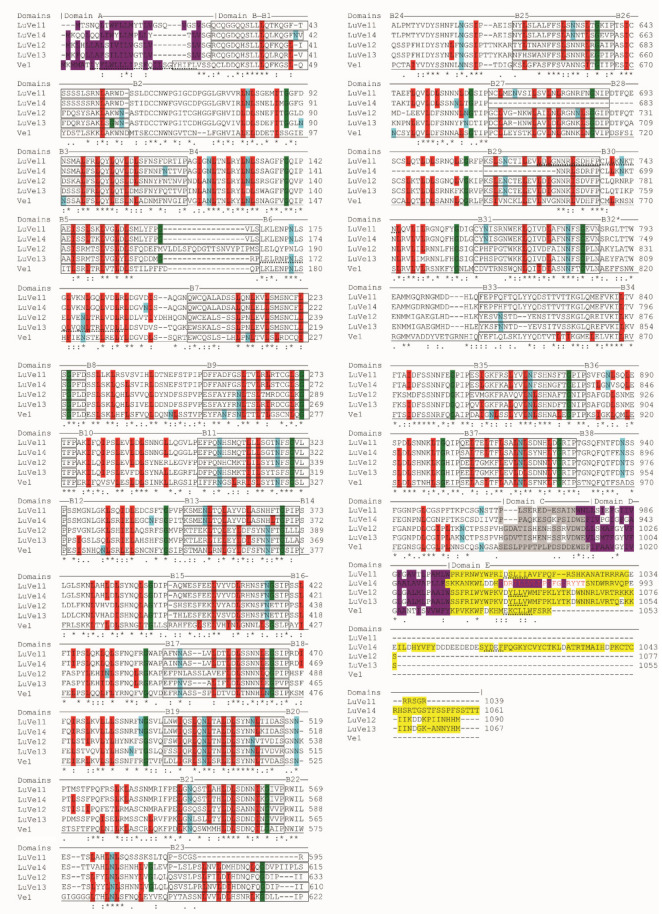
Sequences alignment of the four proteins in flax and Ve1. The proteins are divided into 5 domains A-E as described in the text. Within domain B, 38 sub domains B1-B38 are established corresponding to 37 imperfect copies of the consensus motif [XXIXNLXXLXXLXLSXNXLSGXIP] and B32 a non-LRR segment, highlighted by * (B32). Within domain A and D, hydrophobic amino acids of the putative peptide signal and membrane-associated are highlighted in purple. Within domain B, conserved L/I are in red, G in green and potential N-glycosylation sites in light blue. Within domain C, neutral and acidic amino acids are in grey. Within domain E, basic amino acids are in yellow. Dashed lines represent potential leucine zipper-like motif in domain A for Ve1, in domain B for LuVe11 and LuVe13. Dashed lines in domain E represent endocytosis signals. LuVe14 contains a membrane-associated domain within domain E represented by red amino acids, hydrophobic amino acids are in purple.

**Figure 3 plants-10-00162-f003:**
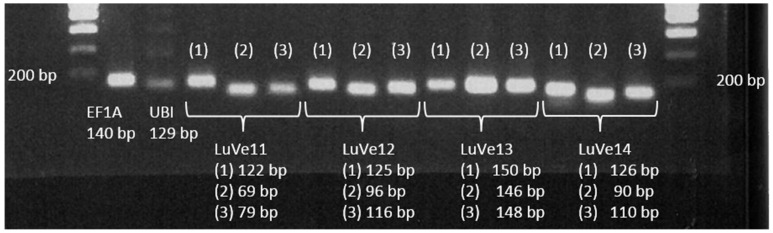
PCR amplifications of EF1A, UBI2, LuVe11 (performed by 3 primer pairs), LuVe12 (performed by 3 primer pairs), LuVe13 (performed by 3 primer pairs), LuVe14 (performed by 3 primer pairs) on cv. Adélie genomic DNA. The respective size of each amplicon is mentioned below.

**Figure 4 plants-10-00162-f004:**
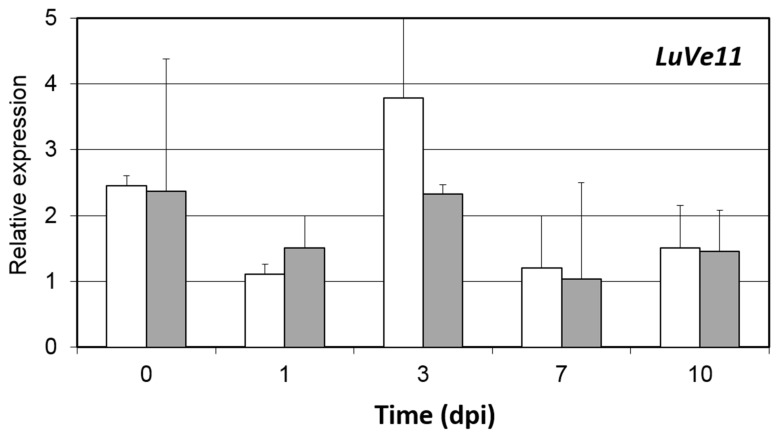
Real-time PCR of a time course of LuVe11 expression in cv. Télïos (white bars) and cv. Adélie (grey bars) in root tissues. Bar represent relative levels of LuVe11 transcripts to the transcript levels of flax EF1A and UBI2 (for normalization) with the SD of a sample of three pooled plants. Dpi, days post inoculation.

**Figure 5 plants-10-00162-f005:**
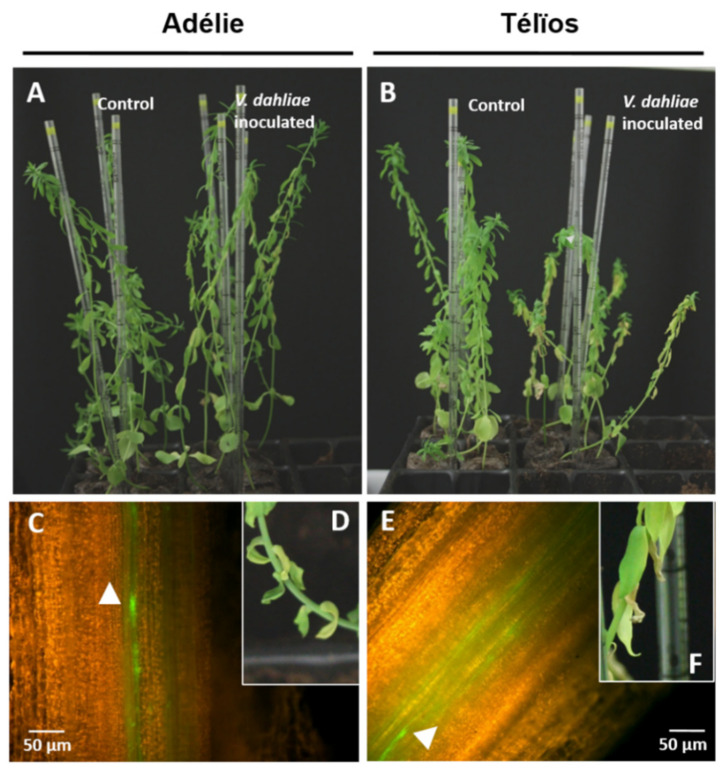
Pathogenicity test performed on two fiber flax cultivars using GFP-tagged V. dahliae race 1 strain. Wilting symptoms observed on cv. Adélie ((**A**) and (**D**), for detailed wilting symptoms on leaves) and cv. Télïos ((**B**) and (**F**), for detailed wilting symptoms on leaves) four weeks post inoculation. Debut of acropetal colonization in infected plant stems, cv. Adélie (**C**) and in cv. Télïos (**E**) at four weeks post inoculation. White arrow shows V. dahliae hyphae.

**Table 1 plants-10-00162-t001:** Description of primers for Ve1 flax homologs and positive controls amplification by PCR.

Gene	Primer Name	Primer Sequence (5′ -> 3′)	Amplicon Size (bp)	Reference
*Ubiquitin extension protein 2*	LuUBI2_1F	5′- CCAAGATCCAGGACAAGGAA -3′	129	[46]
	LuUBI2_1R	5′- GAACCAGGTGGAGAGTCGAT -3′		
*Elongation Factor 1-α*	LuEF1A_1F	5′- GATCGCCTGTCAATCTTGGT -3′	140	
	LuEF1A_1R	5′- GCTGCCAACTTCACATCTCA -3′		
*LuVe11*	LuVe11_1F	5′- TTGGAAGCTTGACTGTCCTG -3′	122	This study
	LuVe11_1R	5′- CCCTTGAAGCAATCCGTTAT -3′		
	LuVe11_2F	5′- GGATTCGGTGCAGTTACCTT -3′	69	
	LuVe11_2R	5′- GATCCTTGGCCAGTACCAGT -3′		
	LuVe11_3F	5′- TGAAACTTCGGTCTGTCTCG -3′	79	
	LuVe11_3R	5′- AAATCGGCAAAGAAATCAGG -3′		
*LuVe12*	LuVe12_1F	5′- CTTTCAACGAAACCATCCCT -3′	125	
	LuVe12_1R	5′- TACAGCGAGAGAGGAAGCAA -3′		
	LuVe12_2F	5′- TCGAGGTAAGGCTCCTGAGT -3′	95	
	LuVe12_2R	5′- GAGGGATCGGACCTTGTAGA -3′		
	LuVe12_3F	5′- CCAAGTTCACCAGTTCATGG -3′	116	
	LuVe12_3R	5′- ATAAGAGCGCCTAAGCCAAA -3′		
*LuVe13*	LuVe13_1F	5′- CTGCTTGCATCACTGGACTT -3′	150	
	LuVe13_1R	5′- AGGGATGGTTTCGTTGAAAG -3′		
	LuVe13_2F	5′- GGCAGCCTTAGCCAGTTATC -3′	146	
	LuVe13_2R	5′- AAGTCCAGTGATGCAAGCAG -3′		
	LuVe13_3F	5′- GGAACATTTCCTGCCAAGAT -3′	148	
	LuVe13_3R	5′- CACCGGAGAAGCTTGTGTAA -3′		
*LuVe14*	LuVe14_1F	5′- GCGAGGTGGGATTCAACTAT -3′	126	
	LuVe14_1R	5′- GGAGTTACCAAATCCTCCCA -3′		
	LuVe14_2F	5′- CCAAATTTATGCGGTAACCC -3′	90	
	LuVe14_2R	5′- GATGCTTGGTTTCCCTGATT -3′		
	LuVe14_3F	5′- AATCAGGGAAACCAAGCATC -3′	110	
	LuVe14_3R	5′- CTTGTTTGCCTTCTTGCTGA -3′		

**Table 2 plants-10-00162-t002:** Names of Ve1 homologs identified in flax proteome, gene sequence ID on Phytozome, protein sizes expressed in amino acids and BLAST score.

Protein Name	Gene Sequence ID	Protein Size (in Amino Acids)	Score
LuVe11	Lus10003389.g	1039	879.0
LuVe12	Lus10042239.g	1090	821.2
LuVe13	Lus10026415.g	1067	803.1
LuVe14	Lus10003387.g	1061	774.2

**Table 3 plants-10-00162-t003:** Sequence homology analysis of four proteins in flax proteome with Ve1 and homologs of Ve1 based on identity of the ORF amino acids sequence (%). Percentage of identity between Ve1 and flax proteins was mentioned in bold. ^a^ Genebank accession numbers for proteins: Ve1, ACR33105.1; GbVe, ACJ61469.1; mVe1, ACB99683.1; SlVe1, AAP20229.1 and StVe, AAQ82053.1.

Protein ^a^	Ve1	LuVe11	LuVe12	LuVe13	LuVe14	GbVe	mVe1	SlVe1	StVe	AAT51733.1	ACB99693.1	AES86946.1	BAD68095.1
Ve1		**48.8**	**45.3**	**45.4**	**45.9**	50.1	49.1	87.3	81.2	91.2	49.3	44.8	39.5
LuVe11			56.8	56.3	74.0	50.2	42.8	49.5	48.7	48.7	44.1	27.7	47.2
LuVe12				78.9	53.5	45.9	40.6	44.6	44.0	44.8	44.6	43.3	38.7
LuVe13					46.1	45.6	41.2	45.8	44.7	44.9	41.6	43.6	38.3
LuVe14						49.8	41.4	47.7	47.1	47.4	41.6	42.7	36.3
GbVe							42.6	50.3	50.6	50.5	42.4	46.5	38.4
mVe1								50.1	49.1	48.9	96.5	39.4	37.6
SlVe1									81.4	84.7	50.1	44.5	39.7
StVe										82.2	49.7	45.4	39.1
AAT51733.1											48.8	44.2	39.1
ACB99693.1												39.6	37.8
AES86946.1													36.8

## Data Availability

Not applicable.

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
