# Peer review of "Identification of Tomato Ve1 Homologous Proteins in Flax and Assessment for Race-Specific Resistance in Two Fiber FlaxCultivars against Verticillium dahliae Race 1"

_plants, 2021, doi:10.3390/plants10010162_

Round 1

Reviewer 1 Report

For most plants two Ve gene homologues (Ve1, Ve2) exist and often they are located on the same chromosome, but on opposite strands. The found four homologues of the Ve gene in flax may be actually just two homologues, but due to flax genome duplication (which occurred during its evolution, which should be mentioned in the text), they are in two copies (each). To verify this alignment of more Ve protein sequences should be performed.

The authors showed expression of only LuVe11 gene after infection. In some plants only one of the homologues is responsible for the resistance to the pathogen, thus expression of all LuVe genes should be shown.

The authors performed gene expression analysis in 4 hours post infection (marked as 0 dpi). How the authors know that after 4 hours the infection started? A marker of infection should be checked, like some PR gene (e.g. chitinase, NADPH oxidase C), whose expression starts at the very early time post infection.

The authors claim there are no differences in the susceptibility of the two varieties of flax (line 390), however, the photographic documentation of the disease progression (Fig. 5 A, B) clearly shows something completely opposite (Telios variation after infection looks weaker than the Adelie variation).

Description of the inoculation procedure (2.6) should be more elaborated. Readers not familiar with the method, may be confused, for instance bathing the seedlings in conidia was performed twice for 1 hour (like once for 2 hours?).

Conclusions should be rewritten after considering the above-mentioned comments.

Author Response

For most plants two Ve gene homologues (Ve1, Ve2) exist and often they are located on the same chromosome, but on opposite strands. The found four homologues of the Ve gene in flax may be actually just two homologues, but due to flax genome duplication (which occurred during its evolution, which should be mentioned in the text), they are in two copies (each). To verify this alignment of more Ve protein sequences should be performed.

Ok, this interesting element will be mentioned in the discussion in order to complete the argumentation on the evolutionary relationships between the proteins. Nevertheless, the four proteins are homologues although the flax whole-genome duplication happened. There is only paralogues and orthologues.

In Supplemental data S2, a phylogenic tree using more Ve proteins to fig1 was performed. TThe results shows that Ve11/Ve14 and Ve12/Ve13 are clustered together within a distinct clade that includes GbVe but not tomato Ve1.

The authors showed expression of only LuVe11 gene after infection. In some plants only one of the homologues is responsible for the resistance to the pathogen, thus expression of all LuVe genes should be shown.

Yes. However, none of the flax cultivars tested showed resistance feature as observed in race-specific resistance in other plants.

The authors performed gene expression analysis in 4 hours post infection (marked as 0 dpi). How the authors know that after 4 hours the infection started? A marker of infection should be checked, like some PR gene (e.g. chitinase, NADPH oxidase C), whose expression starts at the very early time post infection.

Using confocal microscopy, Blum al., 2018 showed that the very beginning of infection process was found at 4 hours post infection. In this paper, the inoculation and plant growth method was identically the same.

The authors claim there are no differences in the susceptibility of the two varieties of flax (line 390), however, the photographic documentation of the disease progression (Fig. 5 A, B) clearly shows something completely opposite (Telios variation after infection looks weaker than the Adelie variation).

Susceptibility and tolerance were assessed by the plant breeder in the field. In this article, all experiments were performed in growth chamber and according to the figure 5A and B, the cultivars showed the opposite. However, both cultivars exhibited symptoms and showed pathogen colonization within vascular tissues. On resistant tomato cultivars, the pathogen is stopped at the early stage of infection and plants didn't exhibit symptoms (Frandin et al., 2009). Several more examples go in that way in the literature. Accordingly, it is not possible to say that the flax cultivars tested are resistant (specific race resistance).

Description of the inoculation procedure (2.6) should be more elaborated. Readers not familiar with the method, may be confused, for instance bathing the seedlings in conidia was performed twice for 1 hour (like once for 2 hours?).

Bathing in conidia inoculum is repeated twice (1 hour*2), I will provide more details and refer to Blum et al., (same method).

Reviewer 2 Report

This is a good effort.  I have a concern that there could be differences in cultivar response to the pathogen, and that screening for Ve1 homologous genes assumed that this was the only protein involved in resistance. It would have been valuable if a diverse panel of flax were inoculated and screened, instead of this only tolerant cultivar.

The only recommendation is that authors should acknowledge that only two cultivars were tested, yet other cultivars could be having other resistance mechanisms, and even could have variations in expression of ve1. If they can capture that bottleneck in their design, the paper will be acceptable

Author Response

The only recommendation is that authors should acknowledge that only two cultivars were tested

This paper is a preliminary work, that's why we have assigned the manuscript to a short communication. You are right indeed, more cultivars should be tested for screening a gene-specific resistance. Is it necessary to implement a largest experiment for a short communication ?

Round 2

Reviewer 1 Report

I am still not convinced about the supposition of the authors that the LuVe genes encode only one homologue (Ve1). Phylogenetic tree shows that the homologs are clustered in a distinct clade that include GbVe but not Ve1, however, as the location and orientation of the sequences in the linum genome is analogous to the location of Ve1 and Ve2 in other plant species (including tomato), it still may be that they are two different homologs (Ve1 and Ve2). This is supported by the photos of infected flax plants, where clearly one is more resistant than the other. The authors should mention the possibility that the LuVe homolgs represent two different paralogues.

As for the photos of infected flax plants. The infection process may vary from that occuring in tomato, thus the conclusion based on the fluorescent microscopy photos is not strong enough, especially that clearly, there are difefrences in the photos. Either the selected photos were not representative or the authors should refer to this result in the manuscript.

I do not understand the answer to my previous comment:

"The authors showed expression of only LuVe11 gene after infection. In some plants only one of the homologues is responsible for the resistance to the pathogen, thus expression of all LuVe genes should be shown."

Please show expression of the rest of the homologs (like it was shown for LuVe11).

Author Response

R: I am still not convinced about the supposition of the authors that the LuVe genes encode only one homologue (Ve1). Phylogenetic tree shows that the homologs are clustered in a distinct clade that include GbVe but not Ve1, however, as the location and orientation of the sequences in the linum genome is analogous to the location of Ve1 and Ve2 in other plant species (including tomato), it still may be that they are two different homologs (Ve1 and Ve2). This is supported by the photos of infected flax plants, where clearly one is more resistant than the other. The authors should mention the possibility that the LuVe homolgs represent two different paralogues.

A: Ok, location and orientation of LuVe11/LuVe14 and LuVe12/LuVe13 suggest the same organization to Ve1 and Be2. I will mention it in the text. Our amino acids analysis based on (i) identity suggested that LuVe11 was closer to GbVe and Ve1 and (ii) motifs analysis for LuVe11 and Ve1 similarly went in this direction. According to that, we were only focused on LuVe11 expression. Sure, there is not enough elements to state that LuVe12/13/14 are not involved in a potential resistance, but amino acids analysis revealed a lesser link with "true" Ve1 and GbVe. I will precise it the text (and that LuVe are 2 different paralogues). 

R: As for the photos of infected flax plants. The infection process may vary from that occuring in tomato, thus the conclusion based on the fluorescent microscopy photos is not strong enough, especially that clearly, there are difefrences in the photos. Either the selected photos were not representative or the authors should refer to this result in the manuscript.

A: Sure, there are differences in the photos. However, typical symptoms of Verticillium wilt can be observed on Adélie plants, smaller but present. This is confirmed by the presence of the pathogen in the vascular tissues. If I refer to Fradin 2009 and Vallad 2008, the pathogen is stopped at the roots and/or the plants do not show symptoms. I understand that the infection process can vary from plant to plant. Accordingly, I will mention in the text that further symptom analysis would be necessary to confirm. Please note that we have submitted this article as a preliminary report.

Reviewer 2 Report

Inoculations were conducted on two cultivars (tolerant and susceptible). The fact these were susceptible, despite having homologous gene does not mean that every other cultivar will have a similar phenotype. Some cultivars that were not tested could be resistant, probably they do not have genetic changes in the loci. This is why I am recommending the you acknowledge the caveat of this study (i.e., limited to what you tested). Otherwise, it is common that a presence of a gene may not mean that it is effective?

Author Response

R: Inoculations were conducted on two cultivars (tolerant and susceptible). The fact these were susceptible, despite having homologous gene does not mean that every other cultivar will have a similar phenotype. Some cultivars that were not tested could be resistant, probably they do not have genetic changes in the loci. This is why I am recommending the you acknowledge the caveat of this study (i.e., limited to what you tested).

A: Sure, I agree with you. With almost 10,000 unique genotypes in the world, it is highly likely to find one or more resistant flax genotypes. Additionally, it is highly likely that this resistance could be associated to Ve1 race specific resistance, a gene-for-gene type resistance.

Our article is a preliminary report and only 2 commercial cultivars were tested. A methologoly for screening was impleted. These results did'nt state that an effective resistance does not exist within L. usitatissimum. It seems it was unclear in the text, I will precise it.

R: Otherwise, it is common that a presence of a gene may not mean that it is effective?

A: Yes indeed, the best exemple is tomato Ve1 (Fradin et al., 2009) : cultivars/genotypes such as Craigella GCR26 and MoneyMaker are suceptible to V. dahalie race 1 whereas VFN8 Genomic AF272367, Craigella GCR218, Motelle and VFN8 are resistants. All have a Ve1 gene but differ in nucleotide composition (SNP). 

Round 3

Reviewer 1 Report

OK, I accept your comment and ammendments, however there is one more thing. I would like to see other homolog gene expression. Why was it only shown for LuVe11?

Author Response

OK, I accept your comment and ammendments, however there is one more thing. I would like to see other homolog gene expression. Why was it only shown for LuVe11?

We were only focused on LuVe11 because the bioinformatic analysis revealed that LuVe11 was the best candidate to form a verticillium wilt resistance protein, and thus for the expression analysis:

- Greater aa identity of LuVe11 to Ve1 and other "true" homologs: LuVe11 has the greater in aa identity with Ve1 (48.8 %), GbVe (50.2 %, that is greater than Ve1 to GbVe) and other "true" Ve1 homologs (see table 1). AA content within the other LuVe12/13/14 shown more divergence to "true" Ve1 homologs than LuVe11. This analysis was based on GbVe analysis by Zhang 2011.

- Sequence patterns analysis of LuVe11 showed features close to Ve1: the analysis in Fig. 2 based on Kawchuk 2001 revealed that LuVe11 has (i) all patterns present in Ve1 e.g. LRR consensus motif, non LRR segment, N-glycosylation sites, leucine zipper-like motif and endocytosis signal, whereas LuVe12/13/14 lacked of one or more patterns. Furthermore (ii) the numbers of motifs are close to Ve1. 

I will precise it in the text.

This manuscript is a resubmission of an earlier submission. The following is a list of the peer review reports and author responses from that submission.

Round 1

Reviewer 1 Report

This paper asks whether a specific group of LRR genes in flax provides resistance against an increasing fungal pathogen (Verticillium dahliae) for which current control methods are inadequate.

There are several strengths of the study.  First and most importantly, the authors do an excellent job with their visual presentation of the data.  They present a phylogenetic analysis of the small group of homologous proteins in flax and several other crops and perform a comparative assessment of the five domains in the protein in the different species.  They confirm the presence of the protein the plant lines through PCR and confirm expression through RT-PCR.  They then test resistance of two plant lines to the pathogen but find that both lines are susceptible to the pathogen, contrary to their expectations.

There are several weaknesses of the study as well.  The text states that the plant lines that they selected for their work were chosen because the supplier told them that Telios was susceptible and Adelie was resistant (see Lines 126 and 275).  This is a weak justification for chosing the two lines.  It would help if the authors could find a paper in the literature that has data on the performance of these two varieties.  Alternatively, they could go back to the supplier and ask for the original data.  Second, the gene expression data are from roots (Fig 4) while the pathogen data are shown for shoots (Fig 5).  I would have liked to see the expression data for shoots.  Third, the expression data were at 10 dpi whereas the plant pictures are from 28 days dpi. Fourth, Fig. 5 seems to show that the infected Adelie plants are twice the size of the infected Telios plants.  I look at this figure and it seems to me that if they showed biomass data also, then they would see the pattern that they were told to expect.  Instead they focus on leaf curling and presence of hyphae, which is present in both varieties. It would be unfortunate to conclude that Adelie is not resistant when they data are abiguous.  Finally, the discussion is very short and the first two paragraphs are literature review and only the last paragraph talks about the data. The authors could surely say something more about their data.

Overall, this paper has attractive figures and is thoughtful written and a very honest straightforward presentation.  If the authors can could include the stem biomass data, then I think the results might support their initial hypotheses and may provide a more satisfying conclusion.  

Minor points

Line 22 (and elsewhere) amino acid content

Line 88 Plural of genus is genera

Author Response

Dear reviewer,

[...] There are several weaknesses of the study as well. The text states that the plant lines that they selected for their work were chosen because the supplier told them that Telios was susceptible and Adelie was resistant (see Lines 126 and 275). This is a weak justification for chosing the two lines. It would help if the authors could find a paper in the literature that has data on the performance of these two varieties. Alternatively, they could go back to the supplier and ask for the original data.

A: The two cultivars used for this study are susceptible (Telios) and tolerant (Adélie) in field conditions according to the plant breeder. A tolerant cultivar refers to a cultivar that shows less symptoms as a result comparing to a susceptible cultivar.To our knowledge, there is no Verticillium wilt resistant flax cultivar (no symptom). The choice of selecting Adélie is based on this hypothesis:

  • It is likely that the field soil contains a mixture of the two V. dahliae pathotypes (i.e. race 1 and race 2) and,
  • Adélie could have the race-specific resistance to overcome the race 1 but not the race 2 and thus develops less symptoms.

Second, the gene expression data are from roots (Fig 4) while the pathogen data are shown for shoots (Fig 5). I would have liked to see the expression data for shoots.

A: According to the litterature cited, race-specific resistance operate at early stage of infection i.e. when the pathogen invades root. Accordingly, we analyzed gene expression at early stages of infection (fig 4). In fact, Fig 5 was etablished to verify if race-specific resistance has operated or not. In addition, even if the gene was expressed in shoot, plants can not overcome pathogen progression in shoot and impede symptoms.

Third, the expression data were at 10 dpi whereas the plant pictures are from 28 days dpi.

A: According to Fradin et al., 2009, there is distinct expression pattern between susceptible and resistant tomato lines during the early 10 dpi. Based on this, we decided to analyze samples before 10 dpi. 

Fourth, Fig. 5 seems to show that the infected Adelie plants are twice the size of the infected Telios plants. I look at this figure and it seems to me that if they showed biomass data also, then they would see the pattern that they were told to expect. Instead they focus on leaf curling and presence of hyphae, which is present in both varieties. It would be unfortunate to conclude that Adelie is not resistant when they data are abiguous.

A: That is true, according to figure 5, the two cultivars seem to exhibit different level of symptoms at 4 wpi. However, the objective was not to quantify the level of susceptibility but if symptoms related to pathogen infection appears or not. As mentionned in the introduction: "Incompatible interaction to V. dahliae race 1 pathotypes was detected on several tomato cultivars, thereby preventing disease progression on stem and leave". Similar results were observed with Arabidopsis transformed with Ve1 and resistant lettuce cultivars.

I hope these elements contribute to more transparency of this work and I would modify the text if necessary.

Reviewer 2 Report

The paper "Identification of Tomato Ve1 Homologous Proteins 2 in Flax and Assessment for Race-specific Resistance 3 on Two Fiber Flax Cultivars against Verticillium 4 dahliae Race 1" by Blum et al. represent an interesting contribution to the identification of tomato Ve1 homologs in fiber flax. The authors used novel genomic resources to try to identify some of these homologues. The approach was good, but the success was limited as significant resistance to pathogens was not achieved. Likely there are several other homologs or alleles in flax that are involved in resistance to Verticillium dahliae. This contribution is worth publishing but it is hard to understand why the paper was submitted to Vaccines. This work has nothing to do with vaccines though it is related to plant resistance to pathogens; the paper should be submitted to other MDPI journals such as Agriculture, Agronomy, Biology, Plants or Pathogens. 

General comments.

A careful revision of the English writing is strongly recommended. Some suggestions are indicated below but much more work is needed to generate a high-quality paper.

Bioinformatics analyses need some improvements and better figures can be produced to facilitate the reading of the paper. Secondary structure analyses have to be incorporated.

After retrieving the flax sequences from the proteome using the tomato Ve ACR33105.1 sequence the authors were supposed to BLASTP their flax homologs in GenBank and retrieve the closest relatives from this database. Selected sequences from these close relatives were supposed to be included in the phylogenetic analysis.  The phylogenetic tree presented in Figure 1 shows that flax sequences are part of a cluster (cluster B) that contains proteins involved in resistance to Verticillium dahliae ruling out proteins involved in resistance to other pathogens such as Cladosporium fulvum, Leptosphaeria maculans and Venturia inaequalis. However, more proteins involved in resistance to V. dahliae were supposed to be included in the analysis (in cluster B) or a separate tree was supposed to be generated sorting out true homologs as it sees that there are a lot of paralogs in Linum usitatissimum.

Primary sequences were generally well conducted but a secondary structure analysis is badly required. I suggest using PSIPRED http://bioinf.cs.ucl.ac.uk/psipred to generate a PSIPRED Cartoon and a MEMSAT-SVM. Both have a better graphic representation and significantly help readers to follow the description for lines 220-257.

Discussion section could be written better, and is very short.

Other comments

Line 13. Abstract needs significant rewriting. “ During the last decade, fiber flax (Linum usitatissimum L.) is affected by the soil borne ….”. The reader might think that indeed, only during the last decade flax started to be affected by. V. dahliae. I doubt about this. Maybe the authors should say “is increasingly affected”.

Line 15 Please replaced “governed by ….  and homologous proteins were found in various plant families” with “determined by homologs of tomato Ve1, a leucine-rich repeat (LRR) 15 receptor-like protein (RLP)”.

Lines 16-17. “ Herein, homologs of tomato Ve1 were detected on flax proteome database.” Indicate how many homologs have been identified.

Line 18 “others” should be “other”

Lines 18-19. “Verticillium wilt disease resistance protein”. This alternative name of Ve1 proteins has supposed to be added to the sentence from Line 16-17.

Line 21. “primary sequences” should be “primary structure” Proteins have primary, secondary, tertiary structures, and some quaternary structure.

Line 26. Why “ Consequently”? Please reformulate this sentence.

Line 27. Why “However”?

Suggest something like: “The results of this study indicate that complex approaches including pathogenicity tests, microscopic observations and gene expression should be implemented for assessing race-specific resistance mediated by Ve1 within the large collection of flax genotypes.”

Line 38-39. Change “region Normandy” with “region of Normandy” or “Normandy region”.

Line 56. “Leave” should be leaves (plural of leaf).

Line 85. Replace “mammals” with “mammals-specific”

Line 86. Add a comma after “Generally”

Line 90. Replace “on flax predicted proteome” with “from the flax predicted proteome”.

Lines 89-91. Suggest modifying the sentence to something like: “Firstly, an in silico screening was performed in order to select from the flax predicted proteome homologous proteins to tomato Ve1 (accession ACR33105.1); selection was based on amino acids identity, phylogeny and functional annotations.”

Lines 92-94. Suggest modifying the second half of the sentence to something like: “assessed a few selected flax homologous genes and plant behavior upon infection was analyzed using a GFP-tagged V. dahliae race 1.”

Line 125. Add a comma after “respectively”.

Line 134 Replace “has” with “was”.

Line 137. Remove the comma after “then”.  Replace “cell” with “counting chamber”.

Line 144. Leave a space between numbers and h.

Line 161. Replace “by” with “with a”

Line 193. It is quite disappointing that a 3-point standard curve was used to calculate PCR efficiency. This is a minimum.

Line 213. Replace “others” with “other”

Line 221. Move table 3 in supplementary data and include a prediction of secondary structure. This structure can be referred to using the info from lines 221 to 230.

Line 232. Replace “spotted” with “identified”

Figure 1. The top of clade B containing L. lyopersicum needs to be reprocessed. Italicize scientific (Latin) names in the legend. How many AA residues have been used to generate the phylogenetic tree? Provide alignments as supplementary data.

Line 263 Table 3. Prediction using Phobius? Reformulate “Size is expressed in amino acids”. Size of the various fragments of tomato Ve1 and of flax homologs?

Lines 373-374 Reference 3; lines 400-402, reference 15; lines 411-412, reference19, and a few other citations: All the words in titles are capitalized. Please change.

Lines 383-387. References 7 and 8 are in bold. Please change.

Author Response

Dear reviewer,

1. Scope : [...] This work has nothing to do with vaccines though it is related to plant resistance to pathogens; the paper should be submitted to other MDPI journals such as Agriculture, Agronomy, Biology, Plants or Pathogens. 

A: This article was submitted to Vaccine special issue "Immune Mechanisms in Plants". We though that article fitted with the scope.

2. Comments and English writing: A careful revision of the English writing is strongly recommended. Some suggestions are indicated below but much more work is needed to generate a high-quality paper.

A: Ok, I will correct using your comments.

3. Phylogenic tree (figure 1)[...] However, more proteins involved in resistance to V. dahliae were supposed to be included in the analysis (in cluster B) or a separate tree was supposed to be generated sorting out true homologs as it sees that there are a lot of paralogs in Linum usitatissimum. 

A: Not sure that if I include more Ve1-like protein sequences I would distinguish true homologs from paralogs. This figure aims to support that the selected proteins belong to Ve1-like proteins clade. The functionnal analysis performed in Figure 2 seems to be better to answer this question.

[..] How many AA residues have been used to generate the phylogenetic tree?

A: All AA residues have been used.

Provide alignments as supplementary data.

A: I will provide it as supplementary data.

4. Secondary structure : [...] Secondary structure analyses have to be incorporated. [...] I suggest using PSIPRED http://bioinf.cs.ucl.ac.uk/psipred to generate a PSIPRED Cartoon and a MEMSAT-SVM. Both have a better graphic representation and significantly help readers to follow the description for lines 220-257.

A: Thanks for the advice. I will use your link to generate a prediction of the secondary structures. That would be helpful to get more informations and compare the sequences. Consequently, I will modify results and discussion. 

6. Bioinformatics analyses need some improvements and better figures can be produced to facilitate the reading of the paper.

A: Let me know which figures could be improved (I will modify Fig 1, what else ?)

7. Table 3: Line 221. Move table 3 in supplementary data

A: I will move it as supplementary data.

8. Discussion section could be written better, and is very short.

A: I will add new elements related to secondary structure analyses

Reviewer 3 Report

Blum et al. identified potential genes related to resistance to fungal infections in fiber flax. The paper is of high relevance due to the major problem caused by V. dahliae in flax crops. The authors identified 4 homologs of leucine-rich repeat receptor-like protein in the flax genome, check their expression in infected plants and correlated two varieties to their resistance to fungal infection. Although the findings are exciting, the data is still too preliminary to be publish. The main issue is that the experiment only show a correlation of the resistance gene expression to the resistance phenotype. More data would be need to really show that these candidates are really resistance proteins. Knockout or overexpressing strains of flax would be ideal, but I'm not as familiar with the genetic tools in this plant. Is there maybe some reporter systems of other plants that can be used to test this?

Minor comments:

1- Figure 2 is too small and hard to read.

2- Figure 4 is missing statistical test.

3- line 278 is missing a space before "10 dpi".

4- line 334 is missing a space before "Moreover".

Author Response

Dear reviewer,

Blum et al. identified potential genes related to resistance to fungal infections in fiber flax. The paper is of high relevance due to the major problem caused by V. dahliae in flax crops. The authors identified 4 homologs of leucine-rich repeat receptor-like protein in the flax genome, check their expression in infected plants and correlated two varieties to their resistance to fungal infection. Although the findings are exciting, the data is still too preliminary to be publish. The main issue is that the experiment only show a correlation of the resistance gene expression to the resistance phenotype. More data would be need to really show that these candidates are really resistance proteins. Knockout or overexpressing strains of flax would be ideal, but I'm not as familiar with the genetic tools in this plant. Is there maybe some reporter systems of other plants that can be used to test this?

I agree with these remarks and on the fact that further experimentations are necessary for a full research paper. Knockout or overexpressing mutant approach was considered with the UGSF team at the University of Lille, who started to work fiber flax mutagenesis. However, technically speaking, this approach on flax fiber is at early stages and presents many difficulties. Based on these elements, your comments, and in accordance with comment from the other reviewer, we propose to format this article into a short communication rather than a research article.

Minor comments:

1- Figure 2 is too small and hard to read.

2- Figure 4 is missing statistical test.

3- line 278 is missing a space before "10 dpi".

4- line 334 is missing a space before "Moreover".

All these elements were modified in the text.

Round 2

Reviewer 2 Report

The revised paper "Identification of Tomato Ve1 Homologous Proteins in Flax and Assessment for Race-specific Resistance on Two Fiber Flax Cultivars against Verticillium dahliae Race 1" by Blum et al. reports an attempt to identify tomato Ve1 homologs in fiber flax.

Major

Unfortunately, the approach failed to identify a relationship between the genes studied and resistance against Verticillium dahliae Race 1. The authors claim that their approach can be used to identify homologs. This is true but their approach is not novel. The paper was submitted to Vaccine special issue "Immune Mechanisms in Plants". However, the paper fails short in providing any novel information on any "Immune Mechanisms in Plants". The secondary structure analysis, provided in the revised MS, reveals very significant differences (low conservation) in aa composition and folding between the candidate flax proteins and Tomato Ve1. It is unlikely that these genes code for true protein homologs as they are part of a large family of genes. Because the results of this study are negative a full paper is not justified; maybe the authors should format their contribution as a short communication.

Minor

English was not thoroughly revised as requested, only the changes indicated by reviewer were addressed. For example "pathogenic test" (line 379) should be "pathogenicity test" or "pathogen test".

An additional phylogenetic tree of the B clade was not performed. However, even from the tree provided we can see that evolution through gene duplication occurred within each species. Therefore, true homologs cannot be identified.

The figures in the supplementary materials should be arranged in a way that can be seen without zooming at 200%.

Author Response

Dear reviewer,

Unfortunately, the approach failed to identify a relationship between the genes studied and resistance against Verticillium dahliae Race 1. The authors claim that their approach can be used to identify homologs. This is true but their approach is not novel. The paper was submitted to Vaccine special issue "Immune Mechanisms in Plants". However, the paper fails short in providing any novel information on any "Immune Mechanisms in Plants". The secondary structure analysis, provided in the revised MS, reveals very significant differences (low conservation) in aa composition and folding between the candidate flax proteins and Tomato Ve1. It is unlikely that these genes code for true protein homologs as they are part of a large family of genes. Because the results of this study are negative a full paper is not justified; maybe the authors should format their contribution as a short communication.

On behalf of all the co-authors, we agree with reviewer to format this article as a short communication.

English was not thoroughly revised as requested, only the changes indicated by reviewer were addressed. For example "pathogenic test" (line 379) should be "pathogenicity test" or "pathogen test".

Ok, so I am going to use the english editing service of MDPI.

An additional phylogenetic tree of the B clade was not performed. However, even from the tree provided we can see that evolution through gene duplication occurred within each species. Therefore, true homologs cannot be identified.

I performed a new phylogenic analysis for the B clade. The figure was added to supplemental data and comments in the text.

The figures in the supplementary materials should be arranged in a way that can be seen without zooming at 200%.

I increased the resolution and included in the text.

Reviewer 3 Report

I am sympathetic with the difficulty of doing more elaborated experiments, but the paper still felt short on showing any convincing evidence that the LuVe1 genes are really involved in resistance to Verticillium infections. The error bars on the qPCR experiment are very large and therefore, do not show any significant difference between the two tested varieties of flax. The other main issue is that just by comparing two varieties is not enough to have any strong evidence of gene-phenotype relationship. If the authors want to go this route, they should considering doing quantitative trait locus (QTL) experiment or genome-wise association studies (GWAS). For those type of experiment it is necessary many more varieties or strains of flax to rule out the function of genomic differences in other genes.

Overall, I still believe that the authors have some very interesting and promising targets, but they are in pre-mature stage to be considered for publication.